# The Reduced Immunogenicity of Zoster Vaccines in CMV-Seropositive Older Adults Correlates with T Cell Imprinting

**DOI:** 10.3390/vaccines13040340

**Published:** 2025-03-22

**Authors:** Adriana Weinberg, Thao Vu, Michael J. Johnson, D. Scott Schmid, Myron J. Levin

**Affiliations:** 1Department of Pediatrics, School of Medicine, Anschutz Medical Campus, University of Colorado, Aurora, CO 80045, USA; michael.j.johnson@cuanschutz.edu (M.J.J.); myron.levin@cuanschutz.edu (M.J.L.); 2Department of Medicine, School of Medicine, Anschutz Medical Campus, University of Colorado, Aurora, CO 80045, USA; 3Department of Pathology, School of Medicine, Anschutz Medical Campus, University of Colorado, Aurora, CO 80045, USA; 4Department of Biostatistics, School of Medicine, Anschutz Medical Campus, University of Colorado, Aurora, CO 80045, USA; thao.3.vu@cuanschutz.edu; 5Department of Molecular, Cellular and Developmental Biology, University of Colorado Boulder, Boulder, CO 80302, USA; scott.schmid@live.com

**Keywords:** cytomegalovirus, zoster vaccines, immune senescence, vaccine immunogenicity

## Abstract

**Background**: Cytomegalovirus (CMV) infection and age impact immune responses to vaccines. The effect of sex remains controversial. We investigated the relationship between cytomegalovirus-seropositivity, age, and sex and the immunogenicity of the recombinant (RZV) and live (ZVL) zoster vaccines in adults ≥50 years of age. **Methods**: Varicella zoster virus (VZV) glycoprotein E (gE)-specific antibody, antibody avidity, and cell-mediated immunity (CMI) were measured pre-vaccination and at regular intervals over 5 years post-vaccination in 80 RZV and 79 ZVL recipients, including 91 cytomegalovirus-seropositive and 90 female participants. **Results**: *Differences associated with CMV-seropositivity*: lower VZV-gE-CMI in RZV recipients after the first dose of vaccine, but no differences after the 2nd dose; lower VZV-gE-specific antibody avidity in ZVL recipients; and more abundant Th1 and senescent T cells (Tsen) and less abundant regulatory (Treg) and tissue-resident memory T cells (Trm). *Differences associated with older age*: lower antibody responses in RZV recipients and lower Th1 cells. *Differences associated with sex*: none for immunogenicity of either vaccine. *Differences associated with T cell subset abundance*: higher Tsens and lower Tregs or Trms were associated with lower post-dose 1 VZV-gE-specific CMI in RZV recipients, and higher Th1s were associated with higher antibody concentrations. **Conclusions**: The correlation of CMV- and age-associated T cell subsets with the immunogenicity of ZVLs and RZVs suggests that T cell imprinting contributes to the effect of age and CMV on vaccine responses.

## 1. Introduction

Cytomegalovirus (CMV) infection has a profound imprinting effect on the immune system, including the expansion of activated and exhausted CD8+ T cells and the inversion of CD4+ to CD8+ T cell ratios [1,2,3,4]. Some of the T cell changes associated with CMV infection have also been reported in older adults, such that CMV infection has been deemed to contribute to immune senescence [1,5,6,7,8].

The relationship between CMV-seropositivity and response to vaccination was studied in the context of influenza and other vaccines, where vaccine responses were decreased in seropositive vaccinees [9,10,11,12]. However, one group of investigators found the immune responses to the influenza vaccine to be higher in CMV-seropositive young adults [13]. CMV infection was also associated with decreased cell-mediated immune (CMI) responses to zoster vaccine live (ZVL) [14].

Age and sex are additional variables that may influence the immunogenicity of vaccines. Typical of immune senescence of older adults is reduced magnitude and duration of antibody and CMI responses to vaccines [15,16,17,18]. The previously recommended ZVL is included among the vaccines that show a prominent effect of age on their immunogenicity [19,20,21]. Studies generated discordant results on the effect of sex on the immunogenicity of vaccines [22,23,24,25].

The currently recommended recombinant zoster vaccine (RZV), which contains the recombinant VZV glycoprotein E (VZV-gE) and a potent adjuvant AS01B, has much higher efficacy and durability than ZVL in older adults, with >90% efficacy in the first 3 years after immunization and >70% at 10 years [26,27,28,29]. The age of RZV recipients had a minimal effect on the efficacy and immunogenicity of RZV in pivotal studies [30,31,32]. The effect of CMV-seropositivity, whose prevalence increases with age [33,34], has not been studied in the context of RZV.

We determined the effect of CMV-seropositivity, in addition to age and sex, on VZV-gE-specific antibody and CMI responses to RZV and ZVL and assessed the role of T cell imprinting as a potential mediator of these effects.

## 2. Methods

### 2.1. Participants and Study Design

This study was conducted using samples and data from a double-blind study of 159 participants ≥ 50 years of age randomized 1:1 to RZV or ZVL (NCT02114333). The primary outcome of the trial was previously reported [35]. The study was approved by the Colorado Multiple Institutional Review Board and all participants signed informed consent. The study enrolled participants in good health except for treated chronic illnesses typical of the age of the vaccinees. All had prior anti-VZV antibodies; none had prior HZ. Exclusions from the study were immune suppression and receipt of blood products within 3 months and vaccines within 4 weeks. Blood was obtained from all participants for immunologic assessment pre-vaccination and at 1, 3, 12, 24, and 60 months post-vaccination.

### 2.2. CMV Antibody Measurements

Antibodies were measured in plasma using the CMV IgG EIA kit (Gold Standard Diagnostics, cat# 01-140) per the manufacturer’s instructions.

### 2.3. Fluorospot Assays

Peripheral blood mononuclear cells (PBMC) were separated from heparinized blood on Ficoll-Hypaque gradients (Sigma-Aldrich, St. Louis, MO, USA) and cryopreserved within 4 h of collection as previously described [35]. PBMC were thawed, rested overnight in a humidified incubator at 37 °C and 5% CO_2_ at 10^6^ PBMC/mL in a growth medium consisting of RPMI 1640 (Thermo Fisher, Grand Island, NY, USA) with L-glutamine (Gemini Bio-Products, West Sacramento, CA, USA), 10% human AB serum (Gemini Bio-Products), 2% HEPES (Thermo Fisher), and 1% penicillin-streptomycin (Gemini Bio-Products), and counted and stimulated for 48h in 96-well dual-color IFN-γ and IL-2 Fluorospot plates (Mabtech, Nacka Strand, Sweden) with pre-optimized amounts of VZV-gE peptide pools (15-mer overlapping by 11-mer; gift from GlaxoSmithKline, Rixensart, Belgiium) in duplicate wells at 250,000 PBMC/well. Unstimulated and phytohemagglutinin A (Sigma Aldrich)-stimulated controls were included. Assays were performed per the manufacturer’s instructions. Results were reported as the mean number of spot-forming cells (SFCs)/10^6^ PBMC in VZV-gE-stimulated wells after subtraction of the SFCs in mock-stimulated controls. An assay control of PBMC from a single leucopack with known performance characteristics was included in each run for reproducibility and validation. Assays were considered valid if the leucopack results were within pre-established bounds.

### 2.4. VZV-gE Antibody Measurements

The VZV-gE ELISA was performed as previously described using recombinant VZV-gE (VZV-rgE), provided by GSK. Briefly, VZV-rgE at 1.0 µg/mL in carbonate buffer was added to 96-well Immulon microtiter ELISA plates (Dynex Technologies, Inc., Chantilly, VA, USA) at 100 µL/well. After 18–24 h incubation at 4 °C, plates were washed and blocked with 5% skim milk in PBS/Tween20. Test plasma samples at 1:50 in blocking solution were added to antigen-coated duplicate wells at 100 µL/well and incubated at room temperature for 35 min. Plates were washed with 0.1% PBS/Tween 20. Goat anti-human IgG-alkaline phosphatase conjugate diluted 1:1000 in blocking solution was added at 100 µL/well and incubated for 30 min. Disodium nitrophenyl phosphate substrate (Sigma-Aldrich) was added for 10 min, after which the reaction was stopped with 3N NaOH per the manufacturer’s instructions. Plates were read on a spectrophotometer at 405 nm. Adjusted OD was calculated as mean test OD minus blank.

### 2.5. Antibody Avidity

Duplicate plates were prepared as described for VZV-gE ELISA. One plate was used for conventional ELISA and the second plate was washed 4 times with PBS/Tween 20 containing 35 mM diethylamine (DEA) before bound antibodies were detected as above. Avidity was expressed as Avidity Index Units (AIU) = mean OD DEA plate/mean OD PBS wash plate × 100.

### 2.6. T Cell Phenotypic Characterization by Flow Cytometry

Thawed PBMCs were rested overnight at 2.5 × 10^6^ cells/mL in growth medium. Brefeldin A (Sigma, 5 µg/mL), Monensin (Sigma, 5 µg/mL), and anti-CD107a (clone H4A3; BD) were added for the last 16h after which PBMCs were washed and incubated with zombie yellow viability stain (Biolegend, San Diego, CA, USA). PBMCs were then washed in 1% BSA (Sigma-Aldrich) in PBS (Thermo-Fisher) (stain buffer) and incubated with antibodies against the surface markers listed in Appendix A. After fixation and permeabilization, intracellular staining was performed with additional antibodies listed in Appendix A. We acquired ≥200,000 events with the Gallios (Beckman Coulter, Indianopolis, IN, USA) instrument and analyzed them using FlowJo v10.10 (BD Biosciences, Franklin Lakes, NJ, USA) software. The gating strategy is shown in Appendix A.

### 2.7. Statistical Analysis

Descriptive statistics were used to characterize the demographics of the two vaccine groups. At each time point following RZV or ZVL administration, separate multivariable linear regression models were fit to estimate differences between CMV-seropositive and CMV-negative participants regarding VZV-gE-specific CMI and antibody responses while adjusting for age and sex. Similarly, for each pre-RZV and pre-ZVL T cell surface marker, we fitted a multivariable linear regression model to estimate the difference in its frequency between CMV-seropositive and CMV-seronegative recipients while controlling for age and sex.

Post-hoc analyses: for each T cell subset significantly associated with CMV serostatus or age in the primary analysis, Pearson correlations were computed between its frequency and CMI and antibody responses, separately for RZV and ZVL participants.

Statistical significance was defined at the 0.05 level. All analyses were performed using R version 4.4.0 statistical software.

## 3. Results

### 3.1. Characteristics of the Study Population

This study used samples from 80 RZV and 79 ZVL recipients including 91 CMV-seropositive participants (Table 1; Appendix A). The demographic characteristics were similar between RZV and ZVL recipients. Among 87 women, 60 (69%) were CMV-seropositive, whereas among 72 men, 31 (43%) were CMV-seropositive (*p* = 0.02). There were no other appreciable differences between CMV-seropositive and CMV-seronegative participants.

### 3.2. Effect of CMV Serostatus, Age, and Sex on VZV-gE-Specific CMI Responses After RZV or ZVL Administration

In ZVL recipients, differences between CMV-seropositive and -seronegative participants did not reach statistical significance. The most pronounced differences were observed at 5 years after immunization when CMV-seropositive participants showed borderline significantly lower IFNg&IL2 double positive (DP) SFCs after subtraction of pre-vaccination SFCs (*p* = 0.09; Figure 1). However, the borderline significance disappeared after adjustment for age and sex (not depicted).

In contrast, compared with CMV-seronegative, CMV-seropositive RZV recipients had significantly or borderline significantly lower IFNg, IL2, and DP SFCs in response to the first dose of vaccine after subtraction of pre-vaccination levels (IL2 *p* = 0.02; IFNg and DP *p* = 0.08 each; Figure 1). A multivariable analysis that was adjusted for age and sex confirmed these results (IL2 *p* = 0.02; IFNg and DP *p* = 0.06 each; not depicted). After the second dose of vaccine and at subsequent time points, the differences in SFCs between CMV-seropositive and -seronegative RZV recipients did not reach statistical significance.

Age and sex did not affect CMI responses to either vaccine.

### 3.3. Effect of CMV Serostatus, Age, and Sex on Antibody Responses to VZV-gE After RZV or ZVL Administration

In RZV recipients, CMV serostatus did not affect the magnitude of the VZV-gE-specific antibody responses. However, antibody responses of RZV recipients significantly decreased with older age at all time points (*p* ≤ 0.01; Figure 2). The increase in antibody avidity after RZV was not affected by CMV serostatus (not depicted) and showed a similar trend for the effect of age observed in the antibody concentrations, but only reached statistical significance at 24 and 60 months post-vaccination (not depicted). Neither antibody concentration nor avidity were affected by sex.

In ZVL recipients, VZV-gE antibody increases after vaccination were not affected by CMV serostatus, age, or sex. However, increases in VZV-gE-specific antibody avidity after ZVL tended to be lower in CMV-seropositive than -seronegative participants (Figure 3). The difference reached statistical significance at 12 months post-immunization (*p* = 0.02). The adjusted analyses confirmed a negative effect of CMV-seropositivity on the increase in avidity at ≥3 months post-vaccination (*p* ≤ 0.06; not depicted). At 3 months post-ZVL, males had lower increases in VZV-gE antibody avidity than females (*p* = 0.01; Appendix A). However, at other time points, there was no indication of an effect of sex on antibody avidity (Appendix A).

### 3.4. Effect of CMV Serostatus, Age, and Sex on Pre-Vaccination T Cell Phenotypes

We analyzed the frequency of Th1, senescent (Tsen), regulatory (Treg), and tissue-resident (Trm) cell subsets pre-vaccination in 60 participants, equally distributed between RZV and ZVL recipients, including 36 CMV-seropositive and 36 female participants (Appendix A). CD4+ and/or CD8+ IFNg+, TNFa+, CXCR3+, and/or CD107a+ Th1 subsets and CD57+ and KLRG1+ Tsen were higher in CMV-seropositive than -seronegative participants (Figure 4). In contrast, CD25+CD127−, TGFb+, PD1+, TIM3+, LAG3+, and/or PD1+ Treg and CLA+ and/or CD103+ Trm were lower in CMV-seropositive than -seronegative participants.

Older age positively correlated with the frequencies of CD8+CLA+ Trm and of CD8+TGFb+ Treg and negatively correlated with CD8+ Th1 and CD4+IL10+ Treg (Figure 4).

Compared to male, female participants had lower CD4+ and/or CD8+ TIM3+, LAG3+ and/or PD1+ and higher CD4+CD25+ and CD4+CD25+CD127-Treg (Figure 4).

### 3.5. Effect of T Cell Subsets Associated with CMV-Seropositivity or Age on the Immunogenicity of RZV and ZVL

To investigate the potential contribution of T cell subsets to the immunogenicity of RZV and ZVL, we performed correlation analyses of the frequencies of T cell subsets with CMI and antibody responses to RZV or ZVL that differed by CMV serostatus or age. Figure 5 shows that CD8+CLA+Trm and CD8+CD25+CD127-Treg, which were lower in CMV-seropositive than -seronegative participants, were positively associated with the increase in VZV-gE CMI after the first dose of RZV. Conversely, the frequency CD8+CD57+ Tsen, which was higher in CMV-seropositive than -seronegative participants, was negatively associated with the magnitude of the CMI responses after the first dose of RZV.

The frequency of Th1 subsets, which decreased with older age, was positively associated with antibody responses in RZV recipients (Figure 5). There were no other correlations between the frequencies of T cell subsets associated with CMV infection or age and immune responses to RZV or ZVL.

## 4. Discussion

Protection against HZ is primarily conferred by Th1 CMI. Our results show that CMV-seropositivity was associated with decreased VZV-gE Th1 CMI responses in RZV recipients after the first dose of vaccine, but no significant differences were observed after the administration of the 2nd dose of vaccine or after ZVL administration. We and others have shown that VZV-gE-specific T cell responses to RZV have the kinetics of a primary response [36]. We further confirmed this observation by showing that RZV recruits more naïve than memory T cells into the CMI response, while ZVL generates recall responses and recruits roughly equal numbers of naïve and memory cells into the VZV-gE-specific response [37,38]. Together, these results suggest that CMV-seropositivity decreased VZV-gE-specific T cell priming in response to RZV, but the effect on recall responses was less pronounced.

The analysis of the T cell subset distribution in CMV-seropositive compared with seronegative participants and the correlation of CMV-associated T cell phenotypes with CMI provided additional immunologic context to our observations. For example, the frequency of Tsens, which were increased by CMV infection, was associated with lower CMI responses. These findings confirm the previously described associations between CMV-seropositivity and frequency of Tsens and between immune senescence and decreased responses to vaccines [17,39]. Moreover, the frequencies of Trms and Tregs, which were lower in CMV-seropositive than CMV-seronegative participants, were associated with higher CMI responses. Trms and Tregs have in common their contribution to tissue integrity and healing [40,41]. Additional studies are needed to elucidate if this property may promote CMI responses to vaccines. Alternatively, the high frequency of Tregs and Trms and the high primary CMI responses to RZV may be independently modulated by third-party immunologic factors.

CMV infection did not affect the magnitude of the antibody responses to VZV-gE in RZV or ZVL recipients. Unlike VZV-gE-CMI, which is typically detected only in 20–30% of VZV-seropositive adults [30,35], antibodies to VZV-gE are present in virtually all VZV-seropositive adults and represent the largest fraction of anti-VZV antibodies [42]. Thus, it is likely that the anti-VZV-gE antibody response to either zoster vaccine was a recall response, which was not affected by CMV infection.

CMV-seropositivity was associated with decreased avidity of anti-VZV-gE antibodies in ZVL recipients. This observation suggests that the quality of the antibodies generated in response to immunization may be modified by the immunologic dysfunctions associated with CMV infection in older adults. However, we did not find significant associations between T cell subsets modulated by CMV infection and avidity. Immunologic dysfunctions induced by CMV infection that we did not study may explain this effect, including B cell abnormalities such as those described by Frasca et al. [9].

We have previously shown an effect of age on antibody responses of RZV recipients [43]. More recently, other investigators also found an effect of age on antibody responses to RZV in people treated with JAK inhibitors [44]. The new information provided in this study is the negative correlation of older age with the frequency of circulating Th1 cells and the positive association of the frequency of Th1 cells with the magnitude of the antibody responses. In our previous study, we showed that VZV-gE-specific Th1s and VZV-gE-specific antibody concentrations after RZV administration were positively correlated [43]. Here we show that the abundance of nonspecific Th1 cells in circulation before vaccination is also associated with the magnitude of the antibody responses to RZV. RZV-generated antibodies are T-dependent, and our observation suggests that increased availability of Th1 cells to support the antibody response may contribute to the magnitude of the response. In addition, B cell dysfunctions associated with older age may also contribute to this effect [45].

This study was hypothesis-generating and had limitations inherent to its design. We did not correct for multiple comparisons to maximize the sensitivity of the analysis. We had T cell phenotypic data available on a subset of participants only. We did not have the appropriate conditions to prove a cause-and-effect relationship underlying the associations that we uncovered. This effort will have to be undertaken in future mechanistic studies.

## 5. Conclusions

CMV infection had a negative effect on primary responses to vaccination but did not affect recall responses. Both CMV and age, but not sex, were associated with specific T cell profiles, which may have contributed to their effect on the immunogenicity of RZVs and ZVLs.

## Figures and Tables

**Figure 1 vaccines-13-00340-f001:**
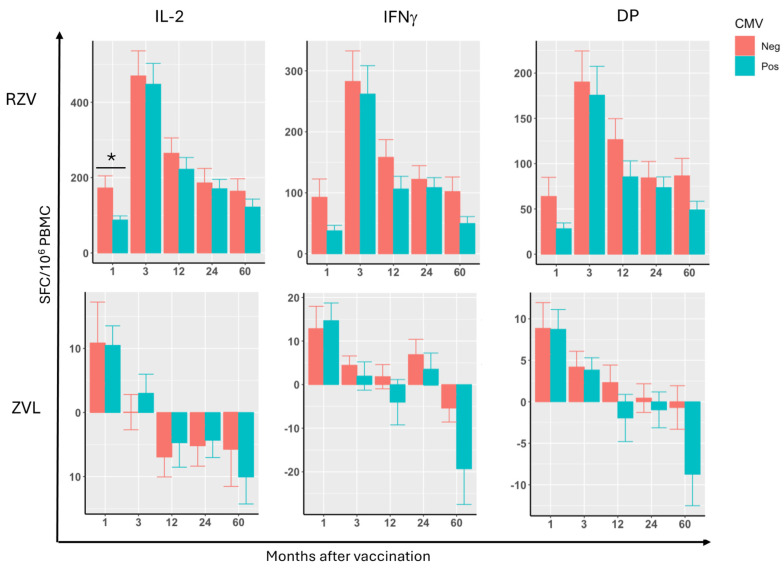
Cell-Mediated Immune Responses to VZV-gE in CMV-Seropositive and -Seronegative RZV and ZVL Recipients. Data were derived from 80 RZV and 79 ZVL vaccinees. Graphs show mean and standard errors of VZV-gE-elicited IL2, IFNg, and IFNg and IL2 double positive (DP) SFCs/10^6^ PBMC as indicated on the graphs in CMV-seropositive (green bars) and -seronegative (red bars) participants at the time points shown on the abscissa after subtraction of pre-vaccination levels. CMV-seropositive RZV recipients (N = 45) had significantly lower IL2 responses (*p* = 0.02) and a trend to lower IFNg and DP responses (*p* = 0.08 for both) than CMV-seronegative recipients after the first dose of vaccine but not at subsequent visits. There were no differences in VZV-gE IL2, IFNg, or DP responses to ZVL between CMV-seropositive (N = 46) and -seronegative vaccinees. The asterisk highlights the comparison that reached statistical significance.

**Figure 2 vaccines-13-00340-f002:**
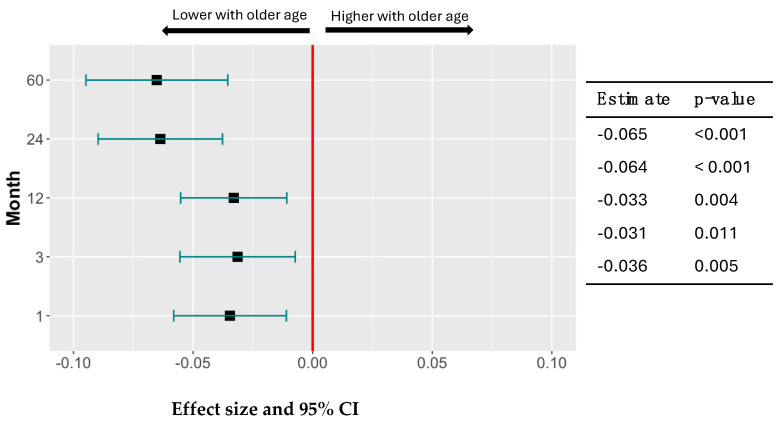
Antibody Responses to gE in RZV Recipients Decrease with Older Age. Data were derived from 80 participants. Forest plots show the effect size and 95% CI of age on the magnitude of VZV-gE-specific antibody increases at each time point after subtraction of pre-vaccination levels. CIs to the left of the red vertical line indicate that antibody responses significantly decreased as age increased. The table shows the point estimates and corresponding *p* values.

**Figure 3 vaccines-13-00340-f003:**
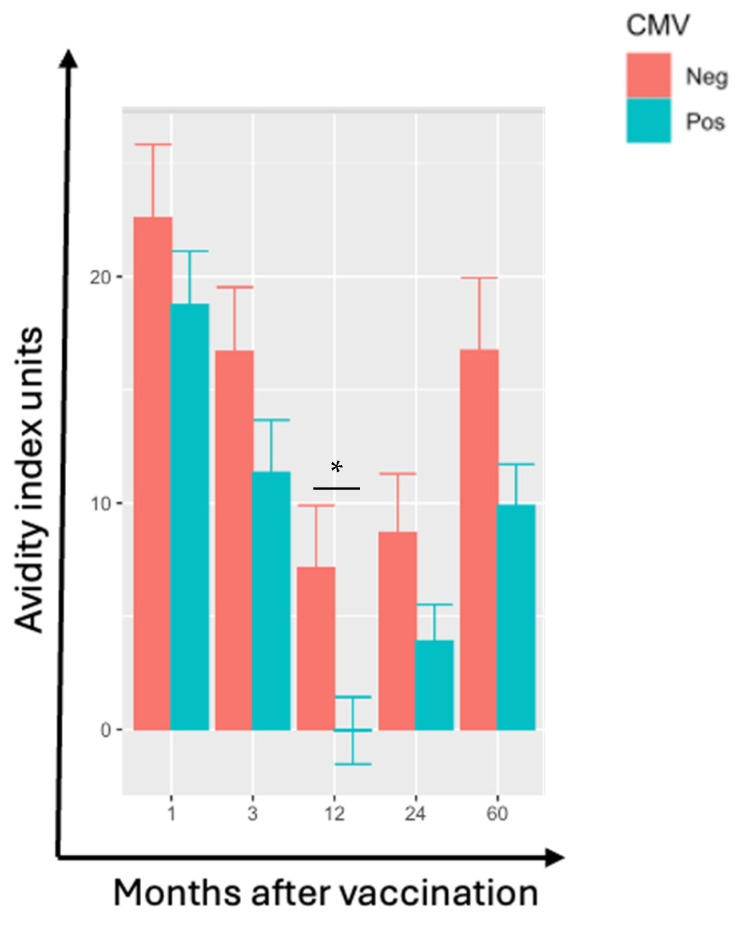
VZV-gE-Specific Antibody Avidity is Lower in CMV-Seropositive than Seronegative ZVL Recipients. Data were derived from 79 ZVL recipients, including 46 CMV-seronegative (red) and 33 CMV-seropositive (green). Bars represent mean and S.E. of the avidity indices at the times after vaccination indicated on the abscissa after subtraction of the avidity indices pre-vaccination. CMV-seropositive participants had lower increases in VZV-gE-specific antibody avidity that were significant at month 12 (*p* = 0.02; asterisk).

**Figure 4 vaccines-13-00340-f004:**
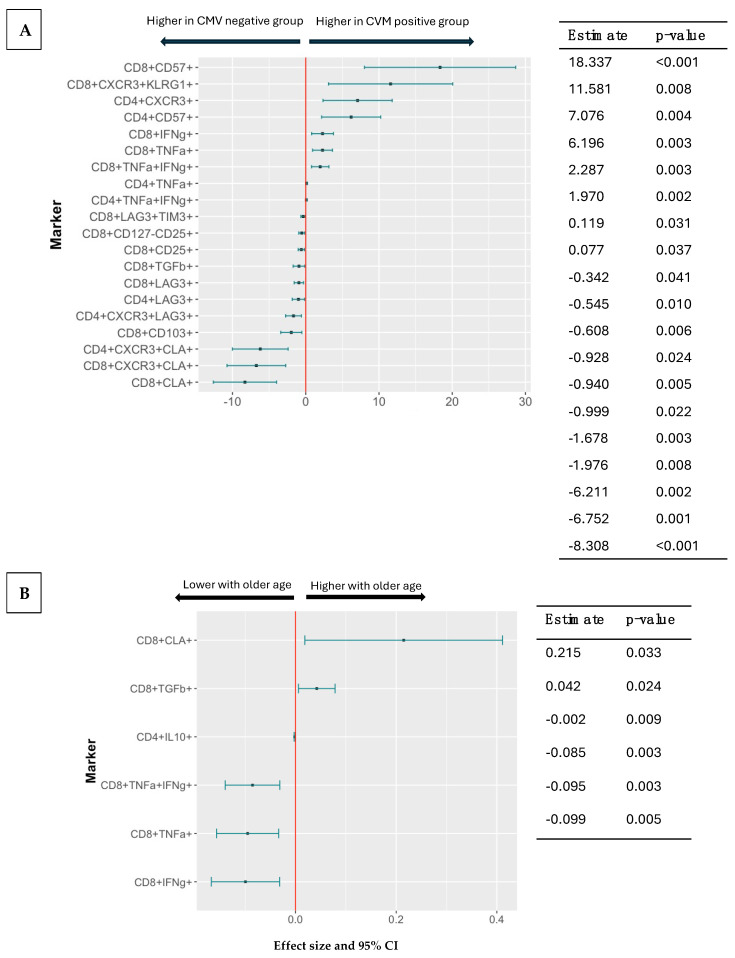
Association of T Cell Subsets with CMV Serostatus, Age, and Sex. Data were derived from 60 participants, 50–85 years of age, including 40 CMV-seropositive and 36 females. The T cell phenotypes were characterized before vaccination by flow cytometry. The forest plots show the strengths of the associations represented by the effect size dot and the 95% CI in the horizontal lines. Point estimates and *p* values are shown in the adjacent tables. (**A**) Significant associations between T cell subset frequencies and CMV serostatus in a multivariate analysis adjusted for age and sex. Markers with 95% CIs to the right side of the red vertical line indicate higher frequencies in CMV-seropositive participants. (**B**) Associations between T cell subset frequencies and age in a multivariate analysis adjusted for CMV status and sex. Markers with CIs to the right side of the red vertical line indicate an increase in the T cell subset frequency as age increases. (**C**) Associations between T cell subset frequencies and sex in a multivariate analysis adjusted for CMV status and age. Markers with 95% CIs to the right of the red vertical line indicate higher frequencies in male participants compared to females.

**Figure 5 vaccines-13-00340-f005:**
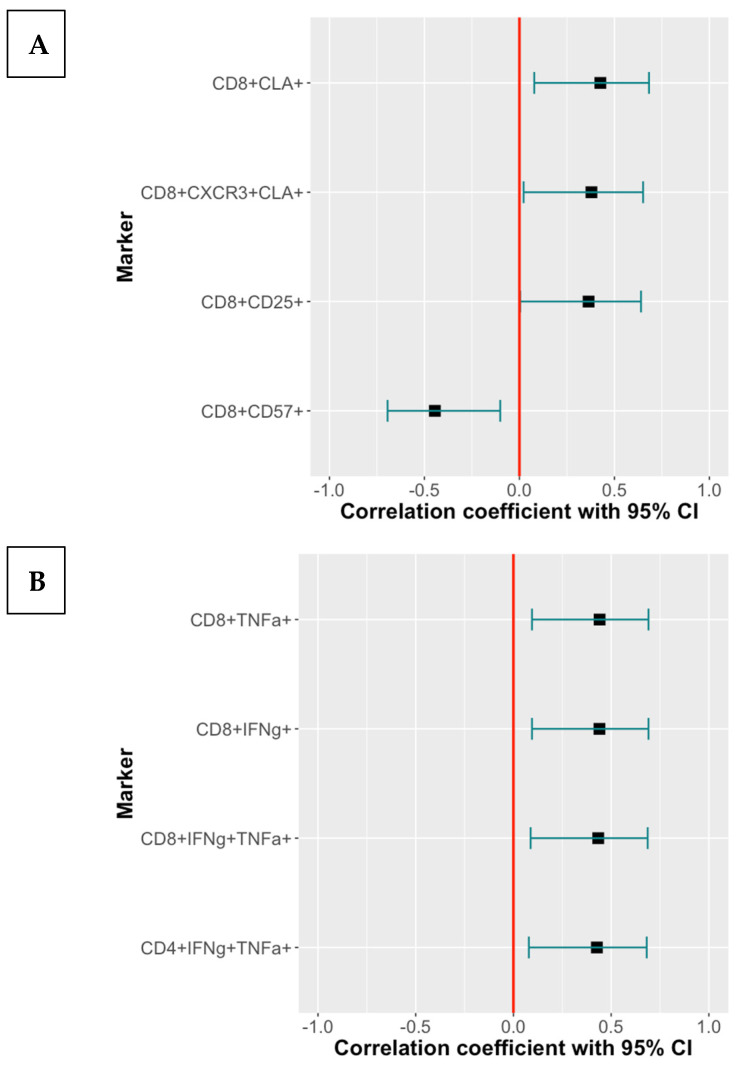
Association of Select T Cell Subsets with CMI and Antibody Responses in RZV Recipients. The data were derived from 30 RZV recipients 56–76 years of age, including 19 CMV-seropositive participants. The forest plots indicate the estimated correlations and their 95% CI between the select T cell subsets significantly associated with VZV-gE-specific IL2 SFCs/10^6^ PBMC after the 1st dose of vaccine (**A**) and with VZV-gE antibody responses at 12 months after vaccination (**B**).

**Table 1 vaccines-13-00340-t001:** Demographic characteristics.

Vaccine		RZV (N = 80)		ZVL (N = 79)	
		CMV-Positive (N = 45)	CMV-Negative (N = 35)	CMV-Positive (N = 46)	CMV-Negative (N = 33)
Median age (IQR)		74 [71, 77]	72 [56, 75]	73 [58, 76]	74 [71, 77]
Sex N (%)	F	28 (62.2)	14 (40.0)	32 (69.6)	16 (62.2)
	M	17 (37.8)	21 (60.0)	14 (30.4)	17 (37.8)
Race N (%)	NW	2 (4.4)	0 (0.0)	2 (4.3)	2 (4.4)
	W	43 (95.6)	35 (100.0)	44 (95.7)	43 (95.6)
Ethnicity N (%)	NH	44 (97.8)	35 (100.0)	44 (95.7)	44 (97.8)
	H	1 (2.2)	0 (0.0)	2 (4.3)	1 (2.2)

Abbreviations: IQR = Inter-quartile range; W = white; NW = non-white; H = Hispanic; NH = non-Hispanic; RZV = recombinant zoster vaccine; ZVL = zoster vaccine live.

## Data Availability

Data are available upon request from Adriana Weinberg after completing a Material Transfer Agreement with the University of Colorado.

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
