# Peer review of "The Reduced Immunogenicity of Zoster Vaccines in CMV-Seropositive Older Adults Correlates with T Cell Imprinting"

_vaccines, 2025, doi:10.3390/vaccines13040340_

Round 1
Reviewer 1 Report
Comments and Suggestions for Authors
In this study, Weinberg, et al. studied the association between T cell responses imprinted by CMV infection in specific cohorts and their responses to shingles vaccination. This is an interesting point and the authors found that CMV sera-status could potentially affect the generation of vaccine response. Also, it is interesting to see that two different types of vaccines (live attenuated one and subunit one) showed different pattern regarding the association between CMV serastatus and Ab avidity. Overall, this is a well-studied project and I have no major comments.
Author Response
In this study, Weinberg, et al. studied the association between T cell responses imprinted by CMV
infection in specific cohorts and their responses to shingles vaccination. This is an interesting point and
the authors found that CMV sera-status could potentially affect the generation of vaccine response. Also,
it is interesting to see that two different types of vaccines (live attenuated one and subunit one) showed
different pattern regarding the association between CMV serastatus and Ab avidity. Overall, this is a well studied project and I have no major comments.
Noted
Reviewer 2 Report
Comments and Suggestions for Authors
In this manuscript, Weinberg et al. explored the immunogenicity of two zoster vaccines (RZV and ZVL) in CMV-seropositive and CMV-seronegative individuals, evaluating VZV-gE antibody titer, avidity, and cell-mediated immune response.
Overall, the authors provided an extensive analysis of the immunization process with both vaccines and effectively discussed the limitations of their study in the discussion section.
However, I have a few suggestions:
- Figure 1: Please modify the Y-axis labeling, as it is not clearly readable.
- Line 74: The authors mention that a recent receipt of blood products or another vaccine was an exclusion criterion. Could they specify the exact timeframe considered as "recent"?
- Figure 4 Caption: It is unclear and should be revised for better clarity.
Author Response
In this manuscript, Weinberg et al. explored the immunogenicity of two zoster vaccines (RZV and ZVL) in
CMV-seropositive and CMV-seronegative individuals, evaluating VZV-gE antibody titer, avidity, and cellmediated immune response.
Overall, the authors provided an extensive analysis of the immunization process with both vaccines and
effectively discussed the limitations of their study in the discussion section.
However, I have a few suggestions:
• Figure 1: Please modify the Y-axis labeling, as it is not clearly readable.
We modified Figure 1 as recommended.
• Line 74: The authors mention that a recent receipt of blood products or another vaccine was an
exclusion criterion. Could they specify the exact timeframe considered as "recent"?
We added the information as requested.
• Figure 4 Caption: It is unclear and should be revised for better clarity.
We revised the caption and added more notes to the figures to improve comprehension. We also
added tables as per Reviewer #3’s request.
Reviewer 3 Report
Comments and Suggestions for Authors
In the manuscript “The reduced immunogenicity of zoster vaccines in CMV-seropositive older adults correlates with T cell imprinting” by Adriana Weinberg and colleagues claim that Cytomegalovirus seropositivity impacts on cell mediated primary immune response to recombinant (RZV) zoster vaccination, whereas live attenuated zoster vaccine (ZVL) results in the production of antigen specific antibodies with low avidity, generation of less regulatory T cells and tissue resident memory T cells. Gender did not interfere with the immune response to both types of vaccines however aged individuals showed lower antibody and Th1 specific responses to RZV.
The manuscript is easy to read and to follow the scientific strategy used to answer the initial question, but there are points to be clarified. To begin, the authors performed a follow-up of the humoral and cellular response two types of zoster vaccines. As reported, the kinetics starts with the time zero (authors claiming that corresponds to the time point before vaccination), later the individuals received a recall dose that is not placed in the kinetics timeline.
This manuscript does not bring innovation to the field.
Points to address:
- Consider to replace the description of the antibodies in material and methods with a supplemental table.
- Figure 1- Statistic values should be inserted in the column plots each time there is a statistic difference.
- Figure 2, 4 and 5 should be supplemented with lateral tables showing the parameters used for the analysis shown in the forest plot.
- Please identify the panels shown in figure 4 and 5.
- View that authors devote several lines on the effect of gender on the antibody response and avidity (lines 226-231) it would be worth it to show the performed analysis as supplemental figure.
- Line 240 please indicate the surface markers used to identify the different T cell subsets, in case give a reference for your choice. In this context is not clear how regulatory T cells have been identified. In the gate strategy, supplementary figure 1, panel 2, authors show the expression of FoxP3 against CD127 and CD25 on gated CD4 T cells but this antibody combination is not plotted in any of the figures; did authors applied it to the study cohort?
- Even if the abbreviations for RZV and ZVL are present in the text I would suggest to add it at the end of table 1
Author Response
In the manuscript “The reduced immunogenicity of zoster vaccines in CMV-seropositive older adults
correlates with T cell imprinting” by Adriana Weinberg and colleagues claim that Cytomegalovirus
seropositivity impacts on cell mediated primary immune response to recombinant (RZV) zoster
vaccination, whereas live attenuated zoster vaccine (ZVL) results in the production of antigen specific
antibodies with low avidity, generation of less regulatory T cells and tissue resident memory T cells.
Gender did not interfere with the immune response to both types of vaccines however aged individuals
showed lower antibody and Th1 specific responses to RZV.
The manuscript is easy to read and to follow the scientific strategy used to answer the initial question, but
there are points to be clarified. To begin, the authors performed a follow-up of the humoral and cellular
response two types of zoster vaccines. As reported, the kinetics starts with the time zero (authors
claiming that corresponds to the time point before vaccination), later the individuals received a recall dose
that is not placed in the kinetics timeline.
This manuscript does not bring innovation to the field.
Points to address:
1. Consider to replace the description of the antibodies in material and methods with a supplemental
table.
We created Table S2 containing the antibodies.
2. Figure 1- Statistic values should be inserted in the column plots each time there is a statistic
difference.
We added asterisks to highlight significant differences.
3. Figure 2, 4 and 5 should be supplemented with lateral tables showing the parameters used for
the analysis shown in the forest plot.
We added tables as recommended.
4. Please identify the panels shown in figure 4 and 5.
We identified the panels
5. View that authors devote several lines on the effect of gender on the antibody response and
avidity (lines 226-231) it would be worth it to show the performed analysis as supplemental figure.
We added a supplemental table.
6. Line 240 please indicate the surface markers used to identify the different T cell subsets, in case
give a reference for your choice. In this context is not clear how regulatory T cells have been
identified. In the gate strategy, supplementary figure 1, panel 2, authors show the expression of
FoxP3 against CD127 and CD25 on gated CD4 T cells but this antibody combination is not
plotted in any of the figures; did authors applied it to the study cohort?
The markers are identified in lines 243-248 of the original manuscript. Lines 324-327 of the
revised manuscript
7. Even if the abbreviations for RZV and ZVL are present in the text I would suggest to add it at the
end of table 1
We added the abbreviations as per reviewer’s request
Round 2
Reviewer 3 Report
Comments and Suggestions for Authors
Authors answered to all the questions I have raised